

# A simple model for the time evolution of the condensation sink in the atmosphere for intermediate Knudsen numbers

Ekaterina Ezhova[1], Veli-Matti Kerminen[1], Kari E. J. Lehtinen[2], and Markku Kulmala[1]

[1]Department of Physics, University of Helsinki, P.O. Box 64, 00014 Helsinki, Finland
[2]Department of Applied Physics, University of Eastern Finland, P.O. Box 1627, 70211 Kuopio, Finland

*Correspondence to:* Ekaterina Ezhova (ekaterina.ezhova@helsinki.fi)

**Abstract.** Transformation of the mass flux towards the particle from the kinetic regime to the continuum regime is often described by the Fuchs-Sutugin coefficient. Kinetic regime can be obtained as a limiting case when only one term of the expansion of the Fuchs-Sutugin coefficient at small $1/Kn$ is considered. Here we take into account the two first terms, and get the mass flux which agrees well with the full mass flux up to $Kn \sim 0.5$. This procedure allows to obtain an analytical

solution of the condensation equation valid for the range of intermediate Knudsen numbers. The expansion is further applied to calculate analytically the condensation sink. The formula for the condensation sink is tested against field observations. The relative contribution of different aerosol modes to the condensation sink is discussed. Furthermore, we present a simple model describing a coupled dynamics of the condensing vapour and the condensation sink. The model gives reasonable predictions of the condensation sink dynamics during the periods of the aerosol modes growth by condensation in the atmosphere.

# 1   Introduction

Condensation sink (CS) is an important parameter for aerosol dynamics quantifying the rate of the vapour condensation on the existing aerosol population. The inverse of CS has a clear physical meaning being the characteristic time scale for vapours to condense onto the surface of existing aerosol. Due to the similarity between the processes of vapour condensation on aerosol particles and coagulation of the smallest particles (monomers, dimers, clusters) with the larger particles from Aitken and

accommodation modes, CS proves useful for the quantification of a coagulation sink (Lehtinen et al., 2007). Then a competition between the process of small clusters coagulating with the larger aerosol particles, represented by CS, and the process of the clusters growth by condensation, represented by particle growth rate, defines the probability of clusters survival and a new particle formation event (Kulmala et al., 2017). A detailed comparative analysis of physical processes in the atmosphere based on the characteristic timescales was performed by Kerminen et al. (2004).

The average condensation sink on new particle formation event days is generally lower as compared with nonevent days (Kulmala et al., 2001; Birmili et al., 2003; Hyvönen et al., 2005; Wu et al., 2007; Asmi et al., 2011; Pikridas et al., 2012; Young et al., 2013; Kanawade et al., 2014; Qi et al., 2015; Vana et al., 2016). Indeed, a large condensation sink means that a relatively large surface of aerosol is available for condensation and coagulation with clusters serving as precursors of newly forming particles. However, at highly polluted places such as megacities, the new particle formation events happen even for





large CS (Wu et al., 2007; Xiao et al., 2015). The dynamics of CS is tightly connected with different atmospheric processes, including the effects of atmospheric chemistry, meteorology and solar radiation. A simple model allowing to describe the dynamics of a condensation sink in the atmosphere could thus be helpful for understanding of new particle burst and cut-off processes. Here we develop a basis for such a model.

For describing aerosol dynamics, we choose a modal approach. This approach treats the whole aerosol population as a sum of modes, and the equations for the first-order moments of the particle size distribution are obtained based of the aerosol general dynamics equation. The number concentration, geometric mean diameter and standard deviation of each mode can be calculated from the moments (Whitby and McMurry, 1997). Assuming a particular type of the particle size distribution (usually lognormal) makes it possible to get a closed system of equations for the moments. This method is not too expensive
computationally, and is at the same time rather accurate (Whitby and McMurry, 1997). A 'pseudomodal' approach has been used to develop fast and efficient aerosol microphysics modules for large-scale atmospheric modelling purposes (Vignati et al., 2004; Stier et al., 2005; Mann et al., 2010; Pringle et al., 2010; Aquila et al., 2011; Zhang et al., 2012; Liu et al., 2016).

      Instead of using the full general dynamics equation, we focus here on one physical process - aerosol growth by condensation - because of its importance for atmospheric aerosol. The model developed here is tested against atmospheric observations
from a remote site Hyytiälä (Finland), representing semi-clean boreal forests in the Northern hemisphere. Typically one can identify two or three modes with the characteristic diameters less than $200\,\mathrm{nm}$ at this site (dal Maso et al., 2008). Both day and night aerosol population behaviour clearly demonstrate patterns typical for condensational growth. On the contrary, another important physical process, coagulation, while a potentially important sink for growing clusters and nanoparticles, affects little to the particle growth rate unless the number concentration of the growing particle populations is very high (Kerminen et al.,
20    2004).

      Aerosol growth by condensation has been extensively investigated theoretically (e.g. Kulmala, 1993; Vesala et al., 1997; Park and Lee, 2000). Besides the simple formulations for the rates of growth involving different physical phenomena at different scales (e.g., Barrett and Clement, 1988), there are models describing the coupled dynamics of the vapour concentration and aerosol distribution applied to the processes in the atmosphere (Clement et al., 2001) and aerosol chambers (Stock, 1987;
Barrett et al., 1992; Wu and Biswas, 1998). Most of these models, however, are still quite complicated.

      The novelty of the present work is that we obtain analytical formulas for the condensation sink and its time evolution in the range of intermediate Knudsen numbers typical for the atmospheric applications. Two regular approaches which can be found in the literature involve either extensive calculations starting with Boltzmann equations (e.g., Kosuge et al., 2010) or employ matching functions (Barrett and Clement, 1988), giving the correct expressions in the molecular and continuum limits and
'something in between' in the transitional regime (unless the full Fuchs-Sutugin coefficient is applied, making further analysis possible only by computational methods). Park and Lee (2000) showed that the latter method (harmonic mean) results in a mass flux quite similar to the one obtained with the full Fuchs-Sutugin coefficient (Fuchs and Sutugin, 1971). They obtained an analytical solution of the condensation equation valid for the whole range of diameters. However, this solution is quite complicated and can not be integrated to get an analytical expression for CS. Here we proceed using the first two terms of the
expansion of the Fuchs-Sutugin coefficient in terms of $1/Kn$ for small $1/Kn$. As can be seen later, this makes it possible to find



analytical formulas for the evolution of the particle size distribution and CS for the intermediate range of Knudsen numbers, while remaining close to those calculated using the original Fuchs-Sutugin coefficient. The expansion is in agreement with the full formula for particle diameters up to $\sim 450$ nm, i.e. for Aitken mode and almost the whole range of sizes typical for the accumulation mode, not only in remote places like a typical boreal forest (Hyytiälä, Finland), but also in megacities, such as Beijing in China (e.g. Liu et al., 2017).

Analysis of the time scales typical for the dynamics of CS and vapour concentration in the atmosphere, allows to use a quasi-stationary approach for the vapour concentration and develop a simple model describing the coupled dynamics of the CS and the condensing vapour in the atmosphere during the periods of aerosol growth by condensation inherent in the atmosphere.

## 2 A theoretical model

### 2.1 The kinetic regime vs the intermediate regime

The equation describing the growth of the aerosol population by condensation is

$$\frac{\partial n_d}{\partial t} = -\frac{\partial (I_d n_d)}{\partial d_p}. \tag{1}$$

Here $n_d = n_d(t, d_p)$ is the number particle distribution, $t$ is time, $d_p$ is the diameter of a particle and $I_d$ is the growth rate. The growth rate can be written as follows (Seinfeld and Pandis, 2016):

$$I_d = \frac{\alpha M_A v_c (p - p_{\mathrm{eq}})}{2 \rho_p R T} \beta_1, \tag{2}$$

where $\alpha$ is the mass accomodation coefficient of the condensing vapour, $M_A$ is its molar mass, $v_c$ is the mean speed of the vapour molecules, $p - p_{\mathrm{eq}}$ is the difference between the vapor pressure far from the particle and the equilibrium vapour pressure, $\rho_p$ is the particle density, $R$ is the universal gas constant and $T$ is temperature. The Fuchs-Sutugin (FS) coefficient, $\beta_1$, connects the mass flux towards a molecule in the kinetic regime with that in the continuum regime:

$$\beta_1 = \frac{4Kn(1 + Kn)}{3\alpha(1 + 0.377 Kn + 1.33 Kn(1 + Kn)/\alpha)}. \tag{3}$$

Here the Knudsen number is $Kn = \frac{2\lambda}{d_p}$ where $\lambda$ is the free mean path of condensing molecules. The solutions of the condensation equation have been extensively investigated (e.g., Clement, 1978) and the method of characteristics has proven to be a useful tool.

In the kinetic regime $Kn \gg 1$, $\beta_1 = 1$, thus $I_d$ does not depend on the particle diameter (and time if the vapour pressure difference is kept constant). The solution of equation (1) in the kinetic regime obtained with the method of characteristics is

$$n_d(d_p, t) = n_{d0}(d_p - I_{d,\mathrm{kin}} t), \tag{4}$$



where $I_{d,\mathrm{kin}} = \dfrac{\alpha M_A v_c (p - p_{\mathrm{eq}})}{2\rho_p RT}$ and $n_{d0}$ is the distribution of particles at initial time $t = 0$. If the mode is initially lognormal with the mean geometric diameter of $d_{p0}$, this solution prescribes the growth of the characteristic diameter of the distribution linearly in time without any change in the shape of the distribution.

In the next order in $(1/Kn)$, one obtains from (3):

$$\beta_1 = 1 - \frac{0.377\alpha}{1.33 Kn}. \tag{5}$$

This function is shown in Fig. 1 together with the full FS coefficient and the kinetic regime approximation, $\beta_1 = 1$. Formula (5) shows a good correspondence with the full formula (3) up to $Kn \approx 0.5$, with the overestimation of the mass flux towards the particles not more than 8%. As (5) is obtained from the kinetic regime formula by account of the term of the next order of smallness, we refer to it further as 'correction'. In this case the growth rate is not constant but depends on the particle diameter:

$$I_d = I_{d,\mathrm{kin}}(1 - \frac{0.377\alpha d_p}{2.66\lambda}), \tag{6}$$

which means that the larger particles grow slower than the smaller particles. This difference leads to the narrowing of the initial distribution with time.

One can introduce a limiting diameter as $1/d_{\mathrm{lim}} = \dfrac{0.377\alpha}{2.66\lambda}$, which corresponds to a zero mass flux towards the particle. Note that for the accommodation coefficient $\alpha = 1$ the limiting diameter is on the order $7\lambda$ and $Kn \approx 0.28$, which is beyond
the range of $Kn \gtrsim 0.5$, where the correction can be applied. Thus, the diameters corresponding to a non-physical zero mass flux will not be considered in the framework of the present model.

The solution of the condensation equation obtained with the method of characteristics for intermediate Knudsen numbers is

$$n_d(d_p, t) = \exp\left(\frac{I_{d,\mathrm{kin}} t}{d_{\mathrm{lim}}}\right) n_{0d}\left(d_{\mathrm{lim}} + (d_p - d_{\mathrm{lim}})\exp\left(\frac{I_{d,\mathrm{kin}} t}{d_{\mathrm{lim}}}\right)\right). \tag{7}$$

We next compare the kinetic limit solution and the 'corrected' solution for the constant growth rate of the particles $I_{d,\mathrm{kin}} = 7$
nm/h, typical for a continental boundary layer during summer time (Yli-Juuti et al., 2011), taking the initial lognormal particle number distribution:

$$n(d_p) = \frac{N_0}{\sqrt{2\pi}d_p \ln\sigma_0} \exp\left(-\frac{\ln^2(d_p/d_{p0})}{2\ln^2\sigma_0}\right). \tag{8}$$

The mean free path for the molecules of sulphuric acid, which can be considered as a typical low-volatile condensing vapour, is $\lambda \approx 120$ nm. Taking the accommodation coefficient as $\alpha = 1$, the mean geometric diameter at $t = 0$ s as $d_{p0} = 20$
nm, particle number concentration as $N_0 = 2000$ cm$^{-3}$ and $\sigma_0 = 1.5$, we obtain the solutions in Fig. 2. As was mentioned earlier, the correction leads to a decrease in the growth rate as compared to the kinetic regime, and the distribution shape





changes in time if the correction is applied. Note that the solution obtained numerically employing the full FS coefficient (shown with symbols in Fig. 2) is in close agreement with that given by eq (7).

The solutions of the condensation equation similar to (4) and (7) can be obtained with the method of characteristics for the vapour pressure varying in time. The growth rate can be written as follows

$$I_d(t) = \frac{\alpha M_A v_c (p(t) - p_{\text{eq}})}{2\rho_p R T} \beta_1 \tag{9}$$

and the solution in the kinetic regime is:

$$n_d(d_p, t) = n_{d0}(d_p - \int_0^t I_{d,\text{kin}}(t')dt'), \tag{10}$$

while applying the correction leads to the solution:

$$n_d(d_p, t) = \exp\left(\frac{\int_0^t I_{d,\text{kin}}(t')dt'}{d_{\text{lim}}}\right) n_{0d}\left(d_{\text{lim}} + (d_p - d_{\text{lim}}) \exp\left(\frac{\int_0^t I_{d,\text{kin}}(t')dt'}{d_{\text{lim}}}\right)\right). \tag{11}$$

## 2.2 Condensation sink

In this Subsection we apply the correction to obtain analytical formulas for the condensation sink. The CS reflects the ability of vapours to condense on the aerosol particles and can be calculated from the formula (e.g. Kulmala et al., 2001):

$$\text{CS} = 2\pi D_v \int_0^{d_{p,\text{max}}} d_p \beta n(d_p) dd_p, \tag{12}$$

where $D_v$ is the diffusion coefficient of the condensing vapour and $\beta = 3\alpha\beta_1/4Kn$.

Assuming that dynamics of an aerosol mode is described by solution (4) and taking the initial lognormal distribution (8), in the kinetic regime the integration of (12) yields:

$$\text{CS}_{\text{kin}} = \frac{2\pi D_v d_{p0}^2 N_0}{\lambda} \frac{\alpha}{2.66}\left(\exp(2\ln^2\sigma_0) + 2\left(\frac{I_{d,\text{kin}}t}{d_{p0}}\right)\exp(0.5\ln^2\sigma_0) + \left(\frac{I_{d,\text{kin}}t}{d_{p0}}\right)^2\right). \tag{13}$$

It can be seen that the CS grows in time as $t^2$, which is not surprising given that the CS is proportional to the surface area available for condensation (i.e., the total surface area of the aerosol population), and the diameter of each particle grows linearly in time.

Accounting for the correction valid for intermediate Knudsen numbers gives $\beta = \frac{\alpha d_p}{2.66\lambda} - \frac{0.377}{4}\left(\frac{\alpha}{1.33}\right)^2\left(\frac{d_p}{\lambda}\right)^2$. Thus, taking the initial lognormal distribution (8) and integrating formula (12) in view of solution (7), we obtain the CS evolution in time:





$$\mathrm{CS_{cor}} = \frac{2\pi D_v d_{p0}^2 N_0}{\lambda} \frac{\alpha}{2.66 \exp\left(\frac{2I_{d,\mathrm{kin}}t}{d_\mathrm{lim}}\right)} \left( e^{2\ln^2 \sigma_0} - 2\frac{d_\mathrm{lim}}{d_{p0}} f(t) e^{0.5\ln^2 \sigma_0} + \left(\frac{d_\mathrm{lim}}{d_{p0}}\right)^2 f^2(t) - \right.$$
$$- \frac{0.377\alpha}{1.33 Kn_0 \exp\left(\frac{I_{d,\mathrm{kin}}t}{d_\mathrm{lim}}\right)} \left( e^{4.5\ln^2 \sigma_0} - 3\left(\frac{d_\mathrm{lim}}{d_{p0}}\right) f(t) e^{2\ln^2 \sigma_0} + \right.$$
$$\left. \left. +3\left(\frac{d_\mathrm{lim}}{d_{p0}}\right)^2 f^2(t) e^{0.5\ln^2 \sigma_0} - \left(\frac{d_\mathrm{lim}}{d_{p0}}\right)^3 f^3(t) \right) \right), \tag{14}$$

where $f(t) = 1 - \exp\left(\frac{\int_0^t I_{d,\mathrm{kin}}(t')dt'}{d_\mathrm{lim}}\right)$, $Kn_0 = 2\lambda/d_{p0}$. Note that one can easily obtain the formulas for the CS similar to

(13) and (14) if the vapour pressure varies in time, using the substitution $I_{d,\mathrm{kin}}t \rightarrow \int_0^t I_{d,\mathrm{kin}}(t')dt'$.

If the parameters of the aerosol population do not depend on time, CS in the kinetic regime can be calculated as follows:

$$\mathrm{CS_{kin,0}} = \frac{2\pi D_v d_{p0}^2 N_0}{\lambda} \frac{\alpha}{2.66} \exp(2\ln^2 \sigma_0), \tag{15}$$

The analogous formula defining CS for the intermediate Knudsen numbers is

$$\mathrm{CS_{cor,0}} = \frac{2\pi D_v d_{p0}^2 N_0}{\lambda} \frac{\alpha}{2.66} \exp(2\ln^2 \sigma_0) \left(1 - \frac{0.377\alpha}{1.33 Kn_0} \exp(2.5\ln^2 \sigma_0)\right) = $$
$$= \mathrm{CS_{kin,0}} \left(1 - \frac{0.377\alpha}{1.33 Kn_0} \exp(2.5\ln^2 \sigma_0)\right). \tag{16}$$

Note the term in brackets, similar to the limiting diameter in the formula for the growth rate (6), but including also the

width of the distribution. Clearly, the CS should not be zero and the largest particles in the distribution, making a significant contribution to the CS, should have Knudsen numbers larger than 0.5 not to introduce errors.

In order to demonstrate the influence of correction (5) on CS, we compare the sinks calculated using formulas (15) and (16). Here we investigate the difference between the kinetic regime and correction, while the difference between the correction and the full FS formula for CS will be addressed in the next Section. The ratio of CSs, $\mathrm{CS_{cor,0}}/\mathrm{CS_{kin,0}}$, as a function of Knudsen

number (based on the mean geometric diameter of the distribution) and $\sigma_0$ is displayed in Fig. 3. The difference between $\beta_0$ and $\beta_1$ in Fig. 1 is no more than 40%. Given that the coefficient $\beta$ appears in the integral (12), the range of parameters corresponding to $\mathrm{CS_{cor,0}}/\mathrm{CS_{kin,0}} \leq 0.6$ should be disregarded. This range, as can be seen from Fig. 3, includes small Knudsen numbers and large $\sigma_0$, i.e. wide distributions with large mean geometric diameters. Obviously, for this range of parameters a part of the distribution is beyond the limits of the correction applicability.

Formulas (15) and (16) can be used in quasi-stationary conditions, when the aerosol distribution at every time moment can be approximated by a lognormal one and the parameters defining the distribution, $d_{p0}$, $\sigma_0$ and $N_0$, change slowly in time. Moreover, CS for several modes can be calculated as a sum of the CSs for each of the modes. Note also that the formulas



obtained in this section are valid for the majority of atmospheric conditions, unless the coagulation process is important and the coagulation term has to be added in the GDE (for example, in highly polluted areas). We use formulas (16) for testing the theory against the atmospheric measurements in the next section.

## 3 Comparison with atmospheric observations. Quasi-stationary conditions and a weakly growing mode.

We first evaluate formula (16) using experimental data for quasi-stationary conditions or weakly growing modes (see examples in Figs 4, 5). The experimental data used for analysis in this Section are from the University of Helsinki SMEAR II station (Hari and Kulmala, 2005). Fig. 4 shows the particle size distribution in Hyytiälä on March 27th, 2014. From 00:30 to 07:00, local time, the aerosol mode remains almost unchanged, with the parameters weakly fluctuating around their mean values, and we refer to these conditions as quasi-stationary. During the daytime, there were growing Aitken and accumulation modes, but we use the same formula (16) and account only for the accumulation mode. As a second example, we consider the particle size distribution in Hyytiälä on 01.06.08 (Fig. 5), again using formula (16) for one mode with larger particles.

To calculate the CS from eq (16), we fitted the measured particle size distribution $dN/d(\log_{10} d_p)$ at different times by the lognormal distribution,

$$\frac{dN}{d(\log_{10} d_p)} = \frac{N_0 \ln(10)}{\sqrt{2\pi} \ln \sigma_0} \exp\left(-\frac{\ln^2(d_p/d_{p0})}{2\ln^2 \sigma_0}\right). \tag{17}$$

The parameters $N_0$, $\sigma_0$ and $d_{p0}$ were obtained using the nonlinear least-squares Marquardt-Levenberg algorithm implemented in Gnuplot (Williams et al., 2013).

The parameters of the particle size distributions on March 27th, 2014 and June 1st, 2008 are summarized in Table 1, and the examples of experimental data fitting by the function (17) are shown in the lower panel of Fig. 5.

The performance of the analytical formula (16) is demonstrated in Figs 4, 5, right panels. The deviation of the theoretically calculated CS from the one obtained from experimental data using full FS coefficient is larger when the tails corresponding to the larger particles are not captured, while smaller particles seem not to be important. These examples illustrate the importance of the largest particles contribution to the CS (Lehtinen et al., 2003).

We next aim to separate the errors introduced by the insufficiently good approximation of the particle number distribution and usage of (5) instead of the full FS formula. We consider the fits of experimental data by the lognormal distribution and calculate the CSs: 1) using full FS coefficient, and 2) using approximations of FS coefficient in the kinetic and intermediate regimes. Fig. 6 shows CS for several days in Hyytiälä in spring and summer, calculated from formulas (15) and (16), versus the full CS, eq (12). The correction results in a $5.5\%$ increase in CS, in accordance with Fig. 1 showing that the the mass flux is overestimated by several percent as compared to the full FS formula. At the same time, the kinetic regime formula leads to overestimates of up to 20-25% for larger values of CS. Note that the Knudsen number corresponding to the mean geometric diameter of the mode is $Kn \approx 5.2$ on March 27th, 2014, while $Kn \approx 2.5$ on June 1st, 2008, and the largest particles taken into account here ($d_p = 500$ nm) have $Kn \approx 0.5$, so that the correction should perform well, as follows from Fig. 1.





Thus, we can conclude that the large difference between the theoretical and experimental CSs in Fig. 5 reflects the error due to approximation.

Overall, when considering the size range $< 500$ nm, the CS calculated using the formula (16) underestimates the CS obtained directly from measurements on average by 10-15%, and even more when the particle number distribution corresponding to larger particles is not captured well (see Fig 4, right panel, at 00.30-02:00 when the aerosol mode characterized by low number concentration but large particle diameters sporadically appears). This is to emphasize that not always one mode is enough for the good representation of the CS, even when the mode seems to be clearly prevailing. Fig. 7 shows an example of such a day. The red curve in Fig. 7 corresponds to the theoretical calculations with one mode having the largest number concentration, while the blue curve shows the CS calculated for the two modes (as a sum, based on the approximation of experimental data by two modes). Clearly, both modes have to be accounted in this case.

One can get an estimate on the contribution of different modes to CS using a parameter map in Fig. 8. This map shows CS calculated using formula (16) as a function of $d_{p0}$ and $N_0$ for the fixed $\sigma_0 = 1.5$. Typical Hyytiälä parameters in spring and summer give CS values between 0.001 and 0.01 s$^{-1}$ (Dada et al., 2017). Modes with relatively small number concentrations ($\sim 100$ cm$^{-3}$) and large characteristic diameters are likely to contribute significantly to CS, and the discrepancy between the measured and theoretically calculated CS for one mode is usually due to not accounting for these large particles. At the same time, the concentrations of smallest particles ($\sim 2-3$ nm) in the cluster mode can be very high in the atmosphere during nucleation events, (up to 10 000 cm$^{-3}$ in Hyytiälä), yet they contribute little to the CS until they grow to sufficiently large diameters ($\sim 20$ nm). Even when the particle number concentrations in the cluster mode (2-3 nm) are as high as $10^5$ cm$^{-3}$ (Kontkanen et al., 2017), their contribution to the typical atmospheric condensation sinks is negligibly small. Note that the maximum characteristic diameter shown here is 350 nm. This is due to the fact that our formula is valid for the particle diameters up to 450-500 nm. Thus, we are not able to draw the conclusions about the contributions of the supermicron modes based on the present theory.

## 4 Dynamics of aerosol mode growing by condensation

A coupled model of aerosol mode growing by condensation includes 2 equations:
1) equation of condensation:

$$\frac{\partial n_d}{\partial t} = -\frac{\partial (I_d n_d)}{\partial d_p}, \tag{18}$$

2) equation describing the time evolution of the vapour concentration:

$$\frac{dC}{dt} = Q(t) - \mathrm{CS} \times (C - C_{\mathrm{eq}}), \tag{19}$$

where $C$ is the vapour concentration, $C_{\mathrm{eq}}$ is the equilibrium vapour concentration and $Q(t)$ is the rate of vapour production. The system is coupled in a sense that the equation for the particle number distribution includes the dependence on the vapour





concentration through the growth rate as $I_d \sim (p - p_{\text{eq}}) \sim (C - C_{\text{eq}})$ and at the same time the equation for the vapour concentration contains the term with CS, proportional to the integral of the number particle distribution. This feature makes is difficult to solve eqs (18) and (19) simultaneously. However, as we showed before (Sec. 2), for the intermediate Knudsen numbers $Kn \leq 0.5$ equation (18) can be integrated and, assuming a lognormal distribution, an analytical formula for the evolution of

the CS can be readily obtained, even for the case of the vapour concentration changing in time.

At the same time, both CS and vapor concentration typically change over times scales considerably longer than 5-17 min...

In what follows we consider for simplicity non-volatile vapours with $C_{\text{eq}} = 0$. We proceed to show that eq (19) can also be significantly simplified for the typical atmospheric values of CS. The lowest values of CS in Hyytiälä in spring and summer are around $0.001 \text{ s}^{-1}$ (Dada et al., 2017), corresponding to the time scale $\tau \sim 1/\text{CS} = 17$ min. Generally, however, values of CS

are higher and the corresponding time scales are smaller than 17 min ($\sim 5$ min for Hyytiala). At the same time, both CS and vapour concentration typically change over times scales considerably longer than 5-17 min (e.g., Petäjä et al., 2009; Kontkanen et al., 2016). This means that the solution of equation (19) relaxes fast (with a time scale $1/\text{CS}$) to the quasy-stationary regime, with

$$C(t) = Q(t)/\text{CS}(t). \tag{20}$$

This formula is often used to get the proxies for vapour concentrations (Petäjä et al., 2009), and a similar expression, with corrections due to the time evolution of $Q(t)$ and $\text{CS}(t)$, has been used by Clement et al. (2001) for the analysis of the particle formation processes in Hyytiälä.

Thus, the system of two differential equations can be reduced to a relatively simple system of two algebraic equations:

1) eq (14) describing the CS evolution for the intermediate Knudsen numbers (or eq (13) in the kinetic regime) using the

substitution $I_{d,\text{kin}} t \rightarrow \int_0^t I_{d,\text{kin}}(t') dt'$,

2) eq (20) for the vapour concentration,

and self-consistent dynamics of the system can be obtained from the simple iterations of this system.

Practically, one can start with some initial growth rate, and then calculate the increase in CS using eq (14) during the short period of time, $\delta t \sim 6 - 10$ min. Then the vapour concentration can be found using eq (20) for the new CS, $\text{CS}(t_0 + \delta t) =$

$\text{CS}_0 + \delta \text{CS}$, at the next time step. The growth rate at this time step can be found as $I(t_0 + \delta t) = \dfrac{I(t_0)}{\text{CS}(t_0 + \delta t)/\text{CS}_0}$. If the production rate of condensing vapour is a function of time, then $I(t_0 + \delta t) = \dfrac{I(t_0)Q(t_0 + \delta t)/Q_0}{\text{CS}(t_0 + \delta t)/\text{CS}_0}$. Further, the increase in the CS is calculated from (14) with the new growth rate, and so on.

This procedure can be readily extended for the two aerosol modes if the CS is taken as the sum of the CSs calculated for each of the modes. The results of the model calculations with two modes is shown in Fig. 9 for one day in Hyytiala (July 24th,

2008). On this day, two periods of condensational growth can be identified in Fig. 7, one during night time (from 0:00 to 7:00) and another during day time (from 12:00 to 18:00).

The only free parameter in the system is the initial growth rate, taken to be 2.6 nm/h for the night, and 12 nm/h for the daytime. The time step was $\delta = 6$ min and the initial parameters for the aerosol modes were taken from the approximation




of the experimental data by the lognormal distribution (night time: $N_{01} = 3130$ cm$^{-3}$, $d_{p01} = 57$ nm, $\sigma_{01} = 1.32$, $N_{02} = 165$ cm$^{-3}$, $d_{p02} = 154$ nm, $\sigma_{02} = 1.24$; daytime: $N_{01} = 3365$ cm$^{-3}$, $d_{p01} = 25$ nm, $\sigma_{01} = 1.54$, $N_{02} = 520$ cm$^{-3}$, $d_{p02} = 93$ nm, $\sigma_{02} = 1.31$).

Even such a simple model gives quite reasonable predictions of the time evolution of CS. At night time the predicted values of the CS are higher, but from Fig. 7 it follows that the concentration of the particles decreases, which is something that we do not capture with the model in its present form. However, the model performs well for the characteristic diameter of the growing mode with the larger number concentration. During the daytime both the evolution of CS and diameter of the growing mode are predicted well assuming the constant value of $Q(t)$, but this assumption can not capture the abrupt stop of the growth of the CS in the evening.

Next, we account for the decrease in the particle number concentration during the night time in the simplest way, assuming $n_d \sim \exp(-t/\tau_{\text{loss}})$. This decay can be associated with an additional term on the r.h.s. of equation (18):

$$\frac{\partial n_d}{\partial t} + \frac{\partial (I_d n_d)}{\partial d_p} = -\frac{n_d}{\tau_{\text{loss}}}, \tag{21}$$

and the solutions (4) and (7) have the same form except for they are multiplied by the factor $\exp(-t/\tau_{\text{loss}})$. It follows then from formula (12) that the CSs are again given by the same formulas (13) and (14) simply multiplied by this factor. The results of the model calculations assuming that the number concentration of particles in the mode with a larger number concentration but smaller characteristic diameter (mode 1) is reduced by 35% and $\tau_{loss} = 7$ h (estimates obtained from experimental data by best fitting the number concentration with an exponentially decaying function) are displayed in Fig. 9 by dotted curves. The particle diameter evolution is affected very little while the CS grows significantly slower and shows now a better agreement with the measurements.

Finally, we comment on the choice of the initial diameter of the mode 20 nm (daytime). For small particle diameters the equation for the number particle distribution should most likely include a diffusion term to account for the widening of the distribution. Starting from the smallest diameters we would end up with a non-physical narrow distribution as a result of time evolution (Lehtinen and Kulmala, 2003). Note that in the present calculations the distribution is narrowing (Fig.2) though in nature it is quite opposite. However, the precise form of a distribution seems not to be important for the CS evolution.

## 5 Conclusions

We have obtained a solution for the condensation equation in the range of intermediate Knudsen numbers (up to $\sim 500$ nm). The solution is based on taking two terms of the expansion for the Fuchs-Sutugin coefficient in terms of $(1/Kn)$ at large $Kn$ and is valid both for constant vapour pressure and (with small modifications) for vapour pressure changing in time.

Based on this solution, we have obtained algebraic formulas describing the dynamics of the condensation sink (CS) in time, assuming an initial lognormal particle number-size distribution. We have tested the formulas against experimental data for quasi-stationary conditions. For the typical parameters of aerosol modes in Hyytiälä (Finland), the correction results in 5.5%





overestimation of CS compared with the calculations using the full Fuchs-Sutugin formula. There is also an overall error due to the approximation of the experimental data by the lognormal distribution, which varies and can be up to 50% when the tail of the distribution corresponding to larger particles is not captured well. This error, however, does not exceed 15% when two aerosol modes are considered.

We confirm the previous results by Lehtinen et al. (2003) that CS is defined mostly by Aitken and accumulation modes with characteristic diameters $\geq 50$ nm and show a diagram allowing to estimate the contribution of different modes to the CS, depending on the characteristic diameter of the mode and particle number concentration. We conclude that for typical atmospheric conditions a cluster mode with the characteristic diameter of about $2 - 3$ nm and a large number concentration of about $10000$ cm$^{-3}$ does not contribute significantly to CS until its mean geometric diameter grows to $\geq 20$ nm.

Note that the difference between CS in the kinetic regime and CS in the intermediate regime can be estimated from Fig. 3. CS in the kinetic regime is proportional to the total surface of aerosol per unit volume. This same quantity appears in the extinction coefficient quantifying aerosol optical depth (Sundström et al., 2015). Thus, one can deduce for what parameters CS is suitable to represent aerosol impact on solar irradiance. For a typical lognormal distributions with $\sigma_0 = 1.5$, CS can be used as a proxy for extinction coefficient if an aerosol modes has a mean geometric diameter less than $\sim 120$ nm ($\mathrm{CS_{cor,0}}/\mathrm{CS_{kin,0}} \geq 0.8$).

Note that these parameters are typical for Hyytiälä where a strong correlation between CS and the extinction coefficient at 550 nm has been demonstrated (Virkkula et al., 2011). This is important to have in mind when choosing parameters for the quantification of biosphere-atmosphere feedback loops.

The differential equation for the vapour concentration was coupled with the equation for the evolution of the particle number distribution to obtain a simple self-consistent model of CS dynamics in the atmosphere. For typical atmospheric values of CS, one can use a quasi-steady state solution for the equation for the vapour concentration, as well as the analytical formula for CS can be used. This model gives reasonable results for the dynamics of CS during the periods of pronounced aerosol growth by condensation for the characteristic diameters of the mode $\geq 20$ nm.

Note that in the framework of this model the characteristic diameter of each mode is permanently growing. In nature, in the case we considered, the growth of diameter is likely to be interrupted by the processes related to meteorology, e.g. morning and evening transitions in the boundary layer.

The model can be extended to investigate dynamics of a cluster/nucleation mode with a characteristic diameter of a few nm in the presence of a base mode and a time-dependent vapour concentration. As we showed, these modes will have only a negligibly small effect on the coupled dynamics of the base mode and condensing vapours while the base mode will define the condensation sink for the smallest particles. The simplest way to include the cluster/nucleation mode is to add into the system considered here a general dynamic equation with a nucleation term and a diffusion term (Seinfeld and Pandis, 2016). For larger particles, the time-dependent vapour production rate and a particle phase chemistry effect can be relevant for future investigations.



*Data availability.* Data measured at the SMEAR II station are available on the following website: http://avaa.tdata.fi/web/smart/. The data are licensed under a Creative Commons 4.0 Attribution (CC BY) license.

*Competing interests.* The authors declare that they have no conflict of interest.

*Acknowledgements.* This work was supported by the Academy of Finland Center of Excellence programme (grant no. 307331), ERC Advanced grant to M. K. (ATM-GTP) and Academy of Finland professor grant to M. K. (no. 302958).





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





| Parameter | Hyytiälä, 27.03.2014 | Hyytiälä, 01.06.2008 | Hyytiälä, 24.07.2008, 1st mode | Hyytiälä, 24.07.2008, 2nd mode |
|---|---|---|---|---|
| $d_{p0}$, nm | $48 \pm 7$ | $106 \pm 40$ | $52 \pm 19$ | $132 \pm 40$ |
| $N_0$, cm$^{-3}$ | $2700 \pm 990$ | $2480 \pm 1600$ | $2970 \pm 1240$ | $380 \pm 250$ |
| $\sigma_0$ | $1.62 \pm 0.09$ | $1.42 \pm 0.12$ | $1.35 \pm 0.10$ m/s | $1.31 \pm 0.11$ |

**Table 1.** Parameters of the lognormal distribution for different days.





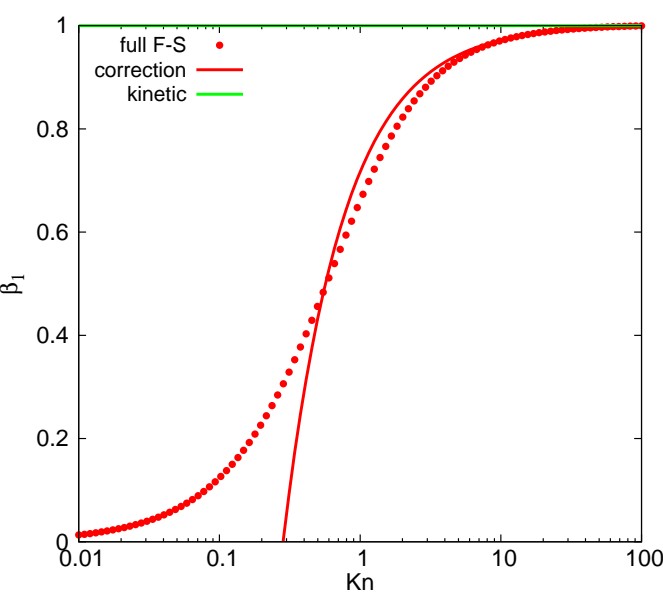

**Figure 1.** Fuchs-Sutugin coefficient as compared to its one term (kinetic) and two terms (correction) expansions at small $1/Kn$.





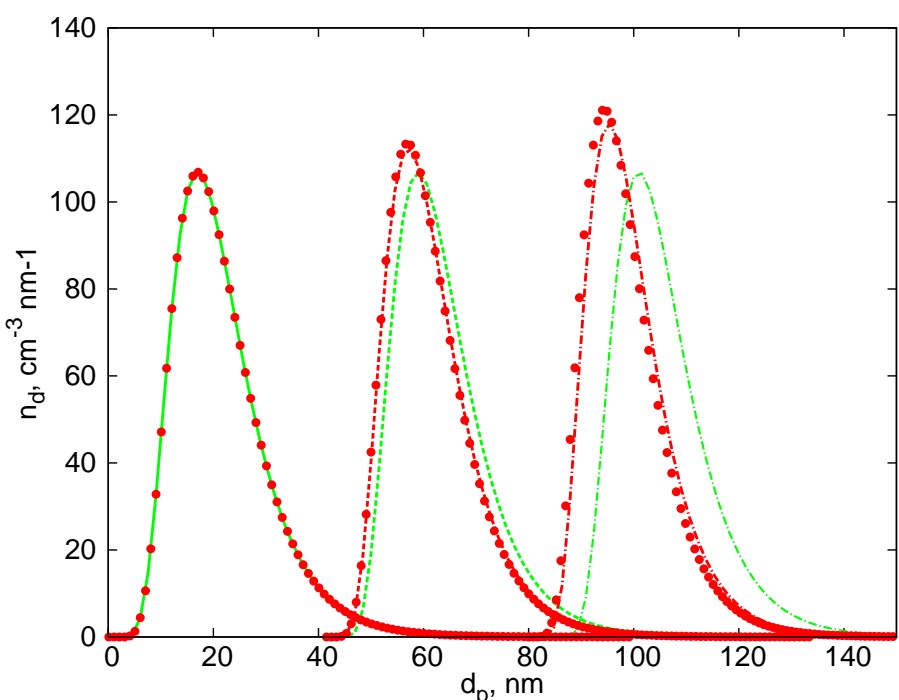

**Figure 2.** Time evolution of the lognormal aerosol distribution: green - solution (4), the kinetic regime; red - solution (7), the solution for intermediate $Kn$. Solid curves - initial distribution, dashed curves - after 6 hours, dash-dotted curves - after 12 hours. Red symbols display the numerical solution of the condensation equation employing the full FS coefficient.



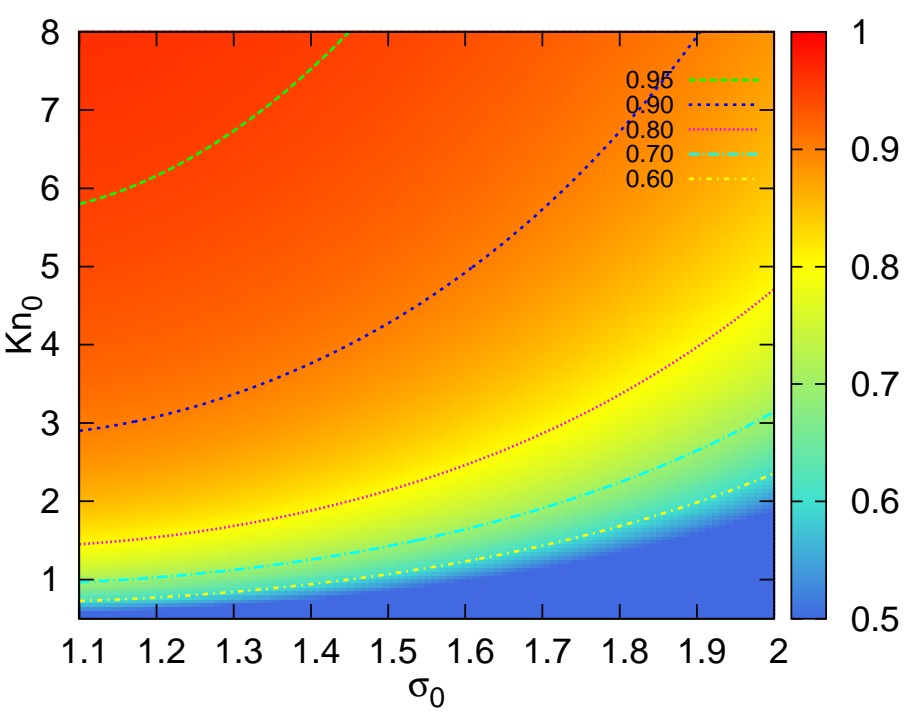

**Figure 3.** The ratio $CS_{cor,0}/CS_{kin,0}$ as a function of $\sigma_0$ and Knudsen number.



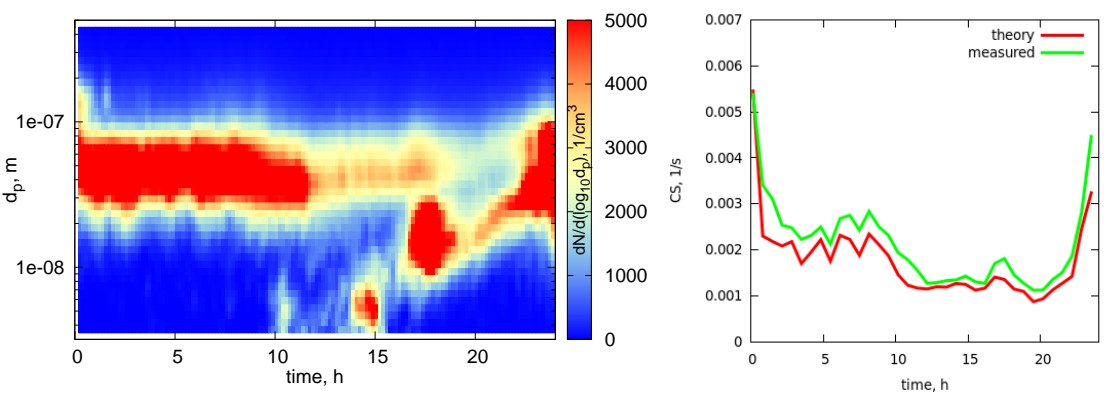

**Figure 4.** Left: Particle size distribution on March 27th, 2014, Hyytiälä. Right: Condensation sink below 500 nm, green: calculated directly from the experimental data, red: calculated using formula (16).



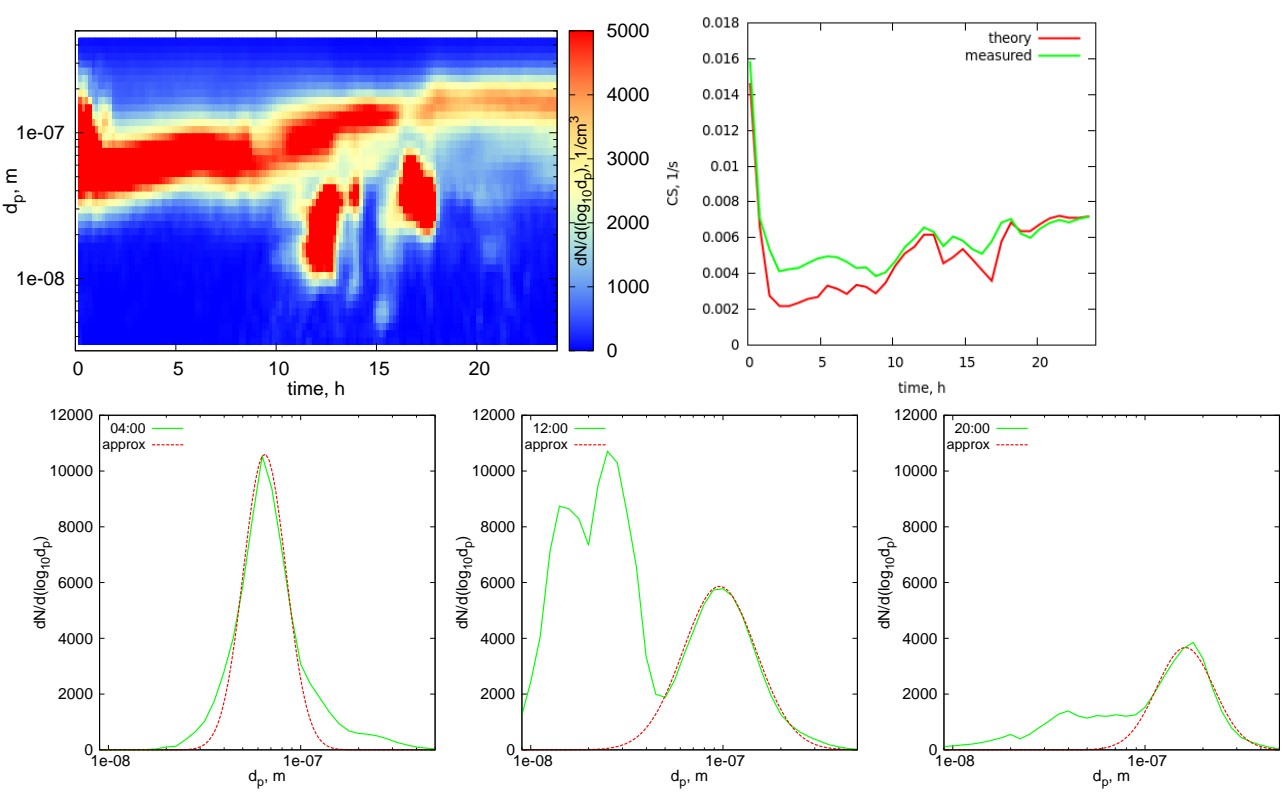

**Figure 5.** Upper panel: (Left) particle size distribution on June 1st, 2008, Hyytiälä. (Right) condensation sink below 500 nm, green: calculated directly from the experimental data, red: calculated using formula (16). Lower panel: examples of one-mode approximation of the particle number distribution.





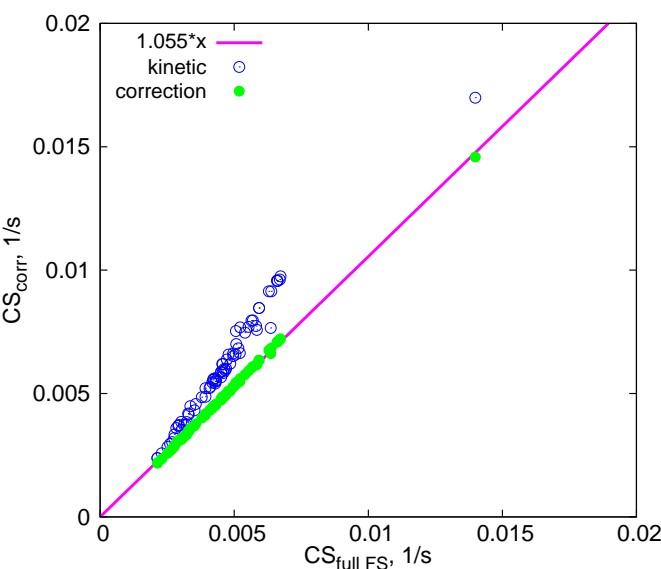

**Figure 6.** CSs calculated using two approximations vs CS calculated using the full FS formula. The violet line corresponds to $CS_{corr} = CS_{fullFS} * 1.055$.





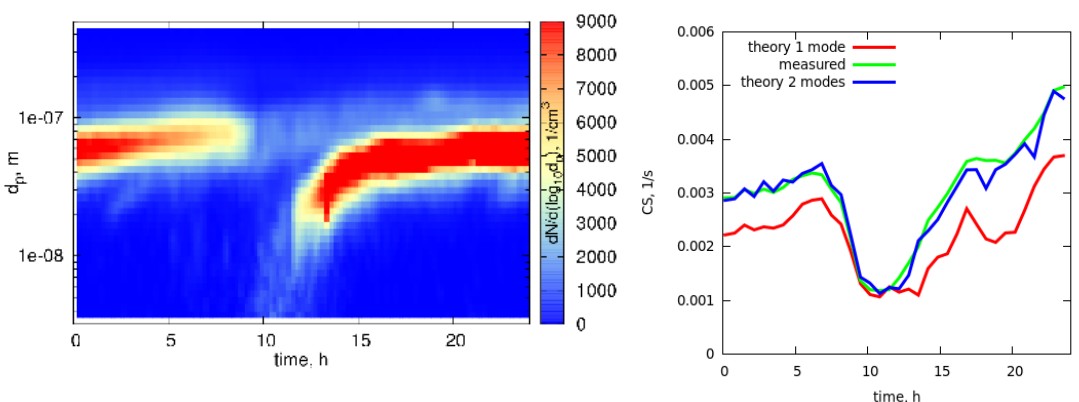

**Figure 7.** Left: Particle size distribution on 24th July 2008. Right: Condensation sink below 500 nm, green: calculated directly from the experimental data, red: calculated using formula (16) for one mode, blue: calculated using formula (16) for two modes.



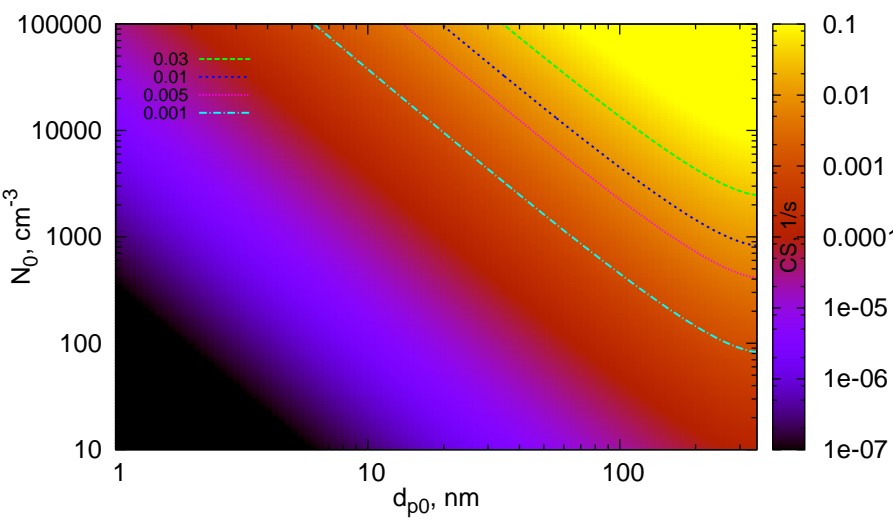

**Figure 8.** A diagram showing the CS as a function of mean geometric diameter and particle number concentration. $\sigma_0 = 1.5$.





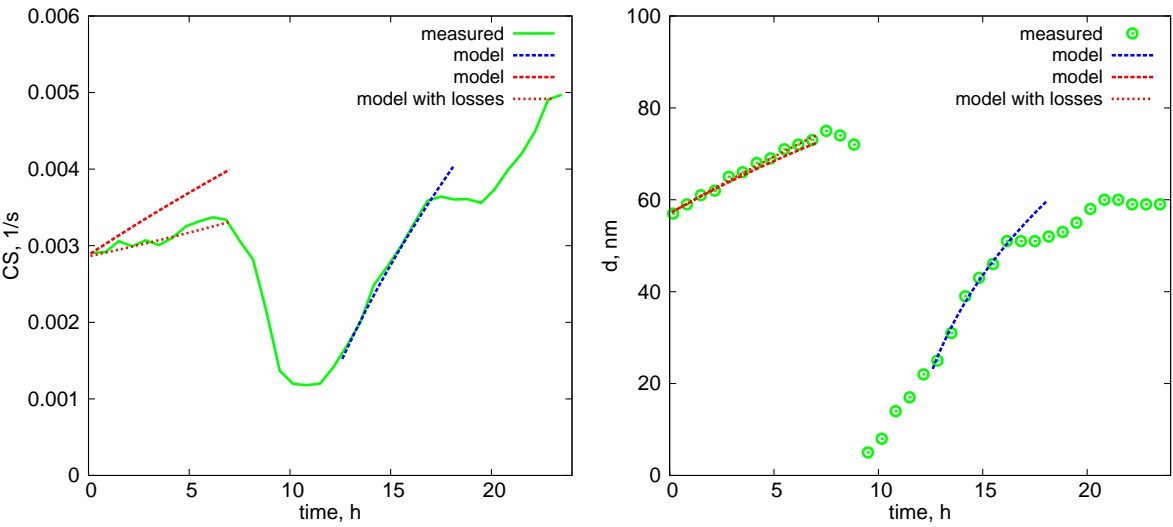

**Figure 9.** Left: Condensation sink as function of time. Green curve: measured (from the approximation of the data by the lognormal distribution), red dashed curve: modelled with the constant particle number concentration, red dotted curve: modelled with the decaying particle number concentration, blue dashed curve: same as for the red dashed curve. Right: Characteristic diameter of the aerosol mode with a larger number concentration as function of time.