# Peer review of "A simple model for the time evolution of the condensation sink in the atmosphere for intermediate Knudsen numbers"

_Atmospheric Chemistry and Physics, 2017_

## Referee Comment (RC1) · Anonymous Referee #1 · 21 Nov 2017

Review: ACP-2017-986

**A simple model for the time evolution of the condensation sink in the atmosphere for intermediate Knudsen numbers**

Ekaterina Ezhova, Veli-Matti Kerminen, Kari E. J. Lehtinen, Markku Kulmala

This paper addresses the accuracy in calculated values of the condensation sink (CS) obtained when using an approximate expression for the Fuchs-Sutugin coefficient (FS). It is shown that this expression, obtained by keeping only the first order term when FS is expanded in terms of $1/Kn$, is sufficiently accurate for $Kn$ values down to 0.5. The authors show that this approximate expression for CS facilitates obtaining an analytic solution to the aerosol general dynamics equation (GDE) by the method of characteristics. This solution accounts for condensation but neglects coagulation, sources, transport, mixing, and other processes that are sometimes important in the atmosphere. The analysis was expanded to include first order losses, leading to an exponential decay in concentration (see Eq. 21). The paper includes results that will be of some value for atmospheric modeling, and was written with reasonable care. It also includes nice mathematical approaches that are not new but are seldom applied. I recommend that the paper be published after the authors consider the following suggestions:

1. Section 2.1 would read better for me if the authors had first shown the solution for time-dependent vapour pressure (solutions given by Equations 10 & 11) followed by the simplified solutions pertinent to constant vapour pressures (Equations 4 & 7). This change is not necessary, but would be an improvement in my view.

2. Figure 9 compares model predictions with observations made on July 24 in Hyytiälä. A first order removal term ($\tau_{loss}$=7 hours) was included in the GDE to account the observed decrease in number concentrations observed at night (see Figure 7). Initial growth rates of 2.6 nm/hour at night and 12 nm/hour in the daytime were input parameters for the model. The authors argue that the model satisfactorily explains the nighttime and daytime observations of CS(t) and $d_p(t)$, but not the large decrease in CS beginning at about 07:00, presumably due to breakup of the inversion after sunrise. While this might be an interesting heuristic result, I question its applicability to the atmosphere. More effort would be needed to demonstrate that all pertinent atmospheric processes have been properly taken into account. For example, does diameter increase linearly and monotonically with time during the daytime on all days at Hyytiälä? If not, it would appear that the authors picked a day with observations that could be explained, not necessarily because the model correctly describes what happened. I would be more comfortable with the paper if Figure 9 were deleted.

3. Figure 3 shows the relationship between CS calculated using the authors' simplified expression and the value of CS calculated using kinetic theory for a range

of aerosol properties pertinent to the atmosphere. The default method for calculating CS for atmospheric data is by using the full FS expression, not the kinetic approach. The only rationale I can see for including this figure has to do with the relationship between the kinetic CS and the extinction coefficient mentioned in the conclusions. As that is not the focus of the paper, I would recommend either that Figure 3 be deleted (my preference) or that it be replaced with a figure that compares the $CS_{corr}$ to $CS_{FS}$.

Minor points:

1. I recommend that in the first sentence following equation (1) the authors explicitly define $n_d=n_d(t,d_p)=dN/dd_p$. Other forms of the distribution function are often used (see, for example, Figures 4, 5, 7 of this paper).

2. Sentence following Equation (5): should be "down to $Kn\approx0.5$", not "up to $Kn\approx0.5$". The approximation is valid in the high $Kn$ (low $1/Kn$) range.

3. Line 27, page 7: "The correction results in a 5.5% increase in CS, in accordance with Fig. 1." I suggest rewording this sentence. Figure 1 shows that for Kn>0.5, the approximate expression for CS exceeds the full expression by an amount ranging from 0 to 8%. So the direction is consistent and the observed 5.5% discrepancy is consistent. That's all we know for certain.

4. p. 8, line 1: "reflects the error due to approximation." The paper addresses several approximations. The text should be clarified to specify which approximation is responsible for the large differences in this case.

5. p. 8, line 5: "(see Fig. 4, right panel, at 00:30 to 02:00, when the aerosol mode characterized by low number concentration but large particle diameters sporadically appears)." I have gone back to look for this several times in Figure 4. It is not apparent to me.

6. p. 9, line 2: "feature makes is difficult" should be " feature makes it difficult"

7. Figures 4, 5 & 7 compare values of CS calculated from "measurements"  (green lines) with those calculated from "theory [i.e., Eq. 16]" (red lines). For clarity the authors should revise this terminology for the following reasons: (1) both of these calculations require the use of measurements, (2) the results calculated directly from measurements (green) are more theoretically correct than those calculated from the simplified model (red). I believe this is shorthand language the authors used to communicate among themselves, but which is not helpful for the reader.

---

## Referee Comment (RC2) · Anonymous Referee #2 · 11 Dec 2017

The manuscript entitled "A simple model for the time evolution of the condensation sink in the atmosphere for intermediate Knudsen numbers" by Ekaterina Ezhova and colleagues describes the effects of an approximated Fuchs-Sutugin (FS) coefficient on the mass flux in the regime of Knudsen numbers down to ∼0.5. The new FS approximation is then used to analyze the impact on a purely calculated condensation sink. The comparison to ambient data from the Hyytiälä measurement site shows agreement within 5.5 % as long as larger particles do not play a big role. The manuscript is well written and clearly fits into the scopes of Atmospheric Chemistry and Physics. The new findings look convincing and novel enough to justify publication in ACP. I have a few minor comments though that should be considered before final publication.

[Figure]

On page 3 just before equ. 3 it is stated that the "...FS coefficient ... connects the mass flux towards a molecule in the kinetic regime...". I believe it should be the "massflux towards a particle".

On page 6, lines 20/21, the authors mention that equ. (15) and (16) can be used in quasi-steady conditions together with distribution parameters changing slowly in time. To get some idea it would be helpful here if corresponding time scales would be mentioned. There is some indication given in a later section but should also be discussed here.

Some clarification is needed in the discussion of Figure 6 on page 7, lines 26-31. Fig. 6 displays CS for several days but then it is referred to "note that Kn $\sim$ 5.2 on March 27th an Kn $\sim$2.5 on June 1st" (line 30). I don't see how this connects to Fig. 6, at least it is not visible to me.

Figure 3: Please expand figure caption, some more explanation is needed here. Also, no reference is made to the color code (condensation sink ratio).

Figure 8: Again, more elaborate figure caption would be desirable. What are the lines representing? BTW: the inset is hard to read.

Figure 9: line types dashed and dotted are hard to distinguish.

Editorial comments: Page 2, lines 18/19: "...affects little the particle growth" (delete "to")

Page 7, line 27: "... showing that the mass..." (delete one "the")

Page 9, line 6: "... time scales..."
* * *

---

## Author Comment (AC1) · 8 Jan 2018

**Replies to the comments of referee 1.**

We are grateful to the referee for the constructive criticism, which helped to improve the clarity of the manuscript. Please find below the replies to the specic comments and an account of the modications implemented.

1. *Section 2.1 would read better for me if the authors had first shown the solution for time-dependent vapour pressure (solutions given by Equations 10 & 11) followed by the simplified solutions pertinent to constant vapour pressures (Equations 4 & 7). This change is not necessary, but would be an improvement in my view.*

We changed Section 2.1 as suggested by the reviewer.

2. *Figure 9 compares model predictions with observations made on July 24 in Hyytiala. A first order removal term ($\tau_{loss}$=7 hours) was included in the GDE to account the observed decrease in number concentrations observed at night (see Figure 7). Initial growth rates of 2.6 nm/hour at night and 12 nm/hour in the daytime were input parameters for the model. The authors argue that the model satisfactorily explains the nighttime and daytime observations of CS(t) and $d_P(t)$, but not the large decrease in CS beginning at about 07:00, presumably due to breakup of the inversion after sunrise. While this might be an interesting heuristic result, I question its applicability to the atmosphere. More effort would be needed to demonstrate that all pertinent atmospheric processes have been properly taken into account. For example, does diameter increase linearly and monotonically with time during the daytime on all days at Hyytiala? If not, it would appear that the authors picked a day with observations that could be explained, not necessarily because the model correctly describes what happened. I would be more comfortable with the paper if Figure 9 were deleted.*

Figure 9 is just an illustration of how the model works in the nearly ideal conditions, when the continuous aerosol growth by condensation can be clearly distinguished in the atmosphere. This holds for mostly sunny days and within one air mass. Generally, aerosol modes do not exhibit this well-pronounced growing dynamics, with the growth process being rather often interrupted either due to the changing air mass, precipitation or some other reason, which must be well understood and parameterized before they can be incorporated into the model. Within the time slots when aerosol mode grows due to condensation, our model can be successfully applied. We do not mean here to state that this model will work perfectly well for any day and any conditions. We would rather keep Figure 9 for illustrative purposes but we additionally stress the applicability of the model to idealized conditions and add this discussion in Lines 7-10, p. 10 and Lines 1-6, p.11.

3. *Figure 3 shows the relationship between CS calculated using the authors' simplified expression and the value of CS calculated using kinetic theory for a range of aerosol properties pertinent to the atmosphere. The default method for calculating CS for atmospheric data is by using the full FS expression, not the kinetic approach. The only rationale I can see for including this figure has to do with the relationship between the kinetic CS and the extinction coefficient mentioned in the conclusions. As that is not the focus of the paper, I would recommend either that Figure 3 be deleted (my preference) or that it be replaced with a figure that compares the $CS_{corr}$ to $CS_{FS}$.*

Using Fig. 3 we wanted to demonstrate, besides the conclusion mentioned by the referee, for what

parameters it is relevant to use the corrected solution as compared to the kinetic regime. In the kinetic regime, as could be seen from our analysis and elsewhere, all the formulas are very simple. It might well be that for some organic vapours, characterized by relatively large molecular weights as compared to sulfuric acid, and, consequently, large mean free paths, the kinetic regime formulas might work reasonably well. Thus, it seems more relevant to compare here the kinetic regime and the correction. That is one of the reasons we would like to keep Fig. 3: in order to give the reader a possibility to estimate the parameters and find out what regime can be used when the chemical formula of the vapour is known (or its molecular weight can be estimated approximately).

Moreover, in spite of the paragraph in the conclusion regarding this figure being indeed not really in the focus of this paper (but still related to this problem), it is still an important note. The question whether CS is a suitable parameter to characterize scatter of light on aerosols is quite important while performing general multidisciplinary analyses on atmosphere-biosphere interaction. The latter usually report the results combining several problems on different topics in one and it would be cumbersome to give all the details in one paper. That is why we would prefer to keep this figure in this paper and to refer to this paper when needed.

Minor points:

1. *I recommend that in the first sentence following equation (1) the authors explicitly define $n_d=n_d(t,d_p)=dN/dd_p$. Other forms of the distribution function are often used (see, for example, Figures 4, 5, 7 of this paper).*

The point is changed as recommended.

2. *Sentence following Equation (5): should be "down to $Kn \approx 0.5$", not "up to $Kn \approx 0.5$". The approximation is valid in the high Kn (low 1/Kn) range.*

The point is changed as recommended.

3. *Line 27, page 7: "The correction results in a 5.5% increase in CS, in accordance with Fig. 1." I suggest rewording this sentence. Figure 1 shows that for $Kn>0.5$, the approximate expression for CS exceeds the full expression by an amount ranging from 0 to 8%. So the direction is consistent and the observed 5.5% discrepancy is consistent. That's all we know for certain.*

We agree with the comment. This sentence now reads: «The correction results in a 5.5% increase in CS, which is consistent with the increase in the mass flux from 0 to 8% as compared to the full FS formula (Fig. 1).»

4. *p. 8, line 1: "reflects the error due to approximation." The paper addresses several approximations. The text should be clarified to specify which approximation is responsible for the large differences in this case.*

We agree with the comment. This sentence is now rephrased to: «reflects the error due to approximating the measured particle number size distribution with a lognormal distribution».

5. *p. 8, line 5: "(see Fig. 4, right panel, at 00:30 to 02:00, when the aerosol mode characterized by low number concentration but large particle diameters sporadically appears)." I have gone back to look for this several times in Figure 4. It is not apparent to me.*

We added the following sentence: «In the left panel this mode is almost not visible due to the non-logarithmic scale of the particle number distribution contour map.»

6. *p. 9, line 2: "feature makes is difficult" should be " feature makes it difficult"*

We changed this in the text.

7. *Figures 4, 5 & 7 compare values of CS calculated from "measurements" (green lines) with those calculated from "theory [i.e., Eq. 16]" (red lines). For clarity the authors should revise this terminology for the following reasons: (1) both of these calculations require the use of measurements, (2) the results calculated directly from measurements (green) are more theoretically correct than those calculated from the simplified model (red). I believe this is shorthand language the authors used to communicate among themselves, but which is not helpful for the reader.*

We agree with the comment. Instead of «measurements» we use now «definition», and instead of «theory» we use «simplified model».

We thank again the referee for the useful suggestions. We hope that you will find that the present manuscript addresses all the comments raised.

---

## Author Comment (AC3) · 8 Jan 2018

**Replies to the comments of referee 2.**

We are grateful to the referee for the constructive criticism, which helped to improve the clarity of the manuscript. Please find below the replies to the specic comments and an account of the modications implemented.

*On page 3 just before equ. 3 it is stated that the "...FS coefficient ... connects the mass flux towards a molecule in the kinetic regime...". I believe it should be the "mass flux towards a particle".*

We agree with the comment and changed this as recommended.

> *On page 6, lines 20/21, the authors mention that equ. (15) and (16) can be used in quasi-steady conditions together with distribution parameters changing slowly in time. To get some idea it would be helpful here if corresponding time scales would be mentioned. There is some indication given in a later section but should also be discussed here.*

We agree with the comment and add the sentence regarding time scales: «...change slowly in time (as discussed later, the typical time scale of the system relaxation to the equilibrium is expected to be not more than 20 min, hence, the time scale on the order of hours or diurnal scale can be used to define the slow change in this context).»

> *Some clarification is needed in the discussion of Figure 6 on page 7, lines 26-31. Fig. 6 displays CS for several days but then it is referred to "note that Kn ~ 5.2 on March 27th and Kn ~ 2.5 on June 1st" (line 30). I don't see how this connects to Fig. 6, at least it is not visible to me.*

We agree with the comment. We changed this sentence to the following: «Note that the Knudsen numbers corresponding to the geometric mean diameters of the modes used for calculations are $Kn >$ 2.5, and the largest particles taken into account here ($d_p$ = 500 nm) have $Kn \sim 0.5$, hence, the correction should perform well, as follows from Fig. 1.»

> *Figure 3: Please expand figure caption, some more explanation is needed here. Also, no reference is made to the color code (condensation sink ratio).*

We agree with the comment. The caption is now expanded to read: «The ratio $CS_{cor,0} = CS_{kin,0}$ as a function of $\sigma_0$ (characterizing the width of the particle number-size distribution) and the Knudsen number $Kn_0$ (corresponding to the mean geometric diameter of the mode).» We added the reference to the color code.

> *Figure 8: Again, more elaborate figure caption would be desirable. What are the lines representing? BTW: the inset is hard to read.*

We changed figure 8 as recommended. The caption is changed to read: «A diagram showing the condensation sink, CS, as a function of the geometric mean diameter of the aerosol mode, $d_{p0}$, and the particle number concentration, $N_0$, for $\sigma_0 = 1.5$.»

> *Figure 9: line types dashed and dotted are hard to distinguish.*

We agree with the comment and changed the line types in figure 9.

**Editorial comments:**

*Page 2, lines 18/19: "...affects little the particle growth" (delete "to")*

We changed this point as recommended.

*Page 7, line 27: "...showing that the mass..." (delete one "the")*

This sentence has been changed due to the comment of the other reviewer.

*Page 9, line 6: "...time scales..."*

We changed this point as recommended.

We thank again the referee for the useful suggestions. We hope that you will find that the present manuscript addresses all the comments raised.